# A Novel Graph-Based Framework for Classifying Urban Functional Zones with Multisource Data and Human Mobility Patterns

**Jifei Wang [2,3], Chen-Chieh Feng [3] and Zhou Guo [1,*]**

[1] School of Geospatial Engineering and Science, Sun Yat-sen University, Zhuhai 519082, China
[2] Department of Geography and Resource Management, The Chinese University of Hong Kong, Hong Kong 999077, China
[3] Department of Geography, National University of Singapore, Singapore 117570, Singapore
[*] Correspondence: guozh37@mail.sysu.edu.cn; Tel.: +86-159-1327-2932

**Abstract:** Recent research has shown the advantages of incorporating multisource geospatial data into the classification of urban functional zones (UFZs), particularly remote sensing and social sensing data. However, the effects of combining datasets of varying quality have not been thoroughly analyzed. In addition, human mobility patterns from social sensing data, which capture signals of human activities, are often represented by origin-destination pairs, thus ignoring spatial relationships between UFZs embedded in mobility trajectories. To address the aforementioned issues, this study proposed a graph-based UFZ classification framework that fuses semantic features from high spatial resolution (HSR) remote sensing images, points of interest, and GPS trajectory data. The framework involves three main steps: (1) High-level scene information in HSR remote sensing imageries was extracted through deep neural networks, and multisource semantic embeddings were constructed based on physical features and social sensing features from multiple geospatial data sources; (2) UFZ mobility graph was constructed by spatially joining trajectory information with UFZs to construct topological connections between functional parcel segments; and (3) UFZ segments and multisource semantic features were transformed into nodes and embeddings in the mobility graphs, and subsequently graph-based models were adopted to identify UFZs. The proposed framework was tested on Zhuhai and Singapore datasets. Results indicated that it outperformed traditional classification methods with an overall accuracy of 76.7% and 84.5% for Zhuhai and Singapore datasets, respectively. The proposed framework contributes to literature in heterogeneous data fusion and is generalizable to other UFZ classification scenarios where human mobility patterns play a role.

**Keywords:** urban functional zone; graph convolutional network; taxi trajectory; location-based service data; remote sensing

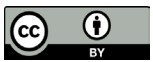

## 1. Introduction

Urban functional zones (UFZs), defined as the irregular urban blocks segmented by roads and carrying meaningful interpretations of urban social functions and economic activities [1], are widely employed as the basic spatial units for studying spatial and social structures of the urban environment [2,3,4] since an ideal UFZ spatial configuration alleviates negative consequences of urbanization, such as traffic congestion, air pollution, and urban heat island [5,6]. To identify UFZs effectively, accurate recognition of their functional layouts is critical [7]. For this task, geographic data describing landscape structures, social sensing data reflecting socioeconomic properties, and the combination of remote and social sensing data describing human mobility, are required to extract information

that captures both the physical and social characteristics of UFZs [8–10]. In terms of extracting the physical characteristics of UFZs, which include landscape compositions and urban morphological structures, high spatial resolution (HSR) remote sensing images have provided detailed information (e.g., spectrums, textures, shapes, angles) of ground objects useful for LULC classification in urban areas and thereby the subsequent UFZ classification [8]. Moreover, the physical features extracted from HSR images are useful for identifying objects of specific land-use types that reflect urban functions, e.g., buildings, forests, and water bodies. Furthermore, the spatial patterns of various land uses are related to the socioeconomic characteristics of individual land parcels. Zhang et al. [7] identified UFZs from a collection of HSR remote sensing images using the convolutional neural network (CNN) model. To achieve UFZ classification in high spatial resolution, Zhou et al. [11] proposed super object convolutional neural network (SO–CNN) to classify UFZs from very high resolution (VHR) remote sensing images. In addition to deep learning methods, Zhang et al. [12] combined bottom-up classification with top-down feedback to improve the results of functional zone mapping based on high-resolution remote sensing images.

However, the visual features from HSR images and static physical representations of individual land use categories do not fully capture urban functions since UFZ is, by definition, a heterogeneous region containing various land use types and diverse objects. Therefore, the location-based social sensing data, such as point of interest (POI) data [11,12], social media check-in data [13,14], mobile phone positioning data [10,15], and GPS trajectory data [16–18], are often used to infer urban functional regions. By integrating extensive social sensing data, the development of data mining methods and GIS analysis techniques enable the discovery of semantic spatial patterns of social functions.

Many attempts have been made to combine heterogeneous data sources for understanding the spatial patterns of urban functions. Using POI data and a simplified Place2vec model, Zhai et al. [19] detected functional regions at a neighborhood scale. Combining POI data with HSR images, Zhang et al. [20] applied hierarchical semantic cognition methods to identify UFZs from HSR images based on POI semantics. By extracting semantic features from topic models, Du et al. [21] realized a large-scale urban functional identification method combining HSR images and POI data. Based on topic models, Tu et al. [22] also extracted social semantics from POIs and integrated them with physical semantics from remote sensing images to perform classifications of urban functional segments at different scales. Nevertheless, data of POIs are often scarce in suburbs and green space regions as opposed to other POI types, resulting in data imbalance issues [21]. In addition, POI data fall short of reflecting dynamic human mobilities since most POIs are names of places or functions of facilities. New social sensing data sources provide more information that represents human mobility and economic activities with high spatial and temporal resolution. Taxi trajectory data [16,23], real-time user data [24,25], and bicycle rental records [26] were all being integrated with POIs to separate functional semantics for different urban structures.

Parallel to adopting disparate data sources for features useful for UFZ classification, recent developments in approaches for UFZ classification explored machine learning models in natural language processing (NLP) to improve UFZ classification accuracy. Du et al. [27] employed topic models and SVM to accurately identify urban functions based on multi-model transportation data, such as POIs, taxi trajectory data, and bicycle rental records. Xu et al. [28] employed ensemble learning and active learning to balance the accuracy of functional zone identification using HSR images and social sensing data. To better understand the correlations between different data sources, Zhang et al. [25] developed a new cross-correlation mechanism to infer urban functions combining HSR image, POI, and real-time user (RTU) data.

These approaches, which focus more on identifying the categories of UFZs based on different classification models, fall short of distinguishing the influence of different com-

binations of metrics on UFZ classification [29–31]. For instance, Tu et al. [32] adopted cluster analysis to integrate landscape metrics with human activity patterns on the grid level in Shenzhen, China, which achieved low accuracy on a large scale UFZ classification and failed to demonstrate the advantages of incorporating human mobility data in UFZ classification. To generalize the relationships between different features critical for UFZs recognition, Xu et al. [28] utilized ensembled models to perform UFZ classification based on multiple metrics from buildings, landscapes, POIs, and human activities. The functional zone recognition of one district in Beijing demonstrated varying importance of multisource metrics; however, the study was limited in spatial scale and model generalization ability. Whilst some research has been carried out on multisource data combinations of UFZ classification, the effectiveness of different types of data has not been fully understood [2,28,33,34].

Another challenging problem in UFZ classification using multiple data sources is the integration of human mobility information. Existing studies [17,23,24] utilizing transportation data have considered only their temporal characteristics when using taxi GPS trajectory data, and most UFZ classification approaches conveniently downplay or neglect the fact that both the spatial transitions and time-series changes must be considered together to properly characterize human mobility. A recent study by Hu et al. [35] applied a graph convolutional neural network (GCNN) approach to taxi trajectory data to identify the functions of road segments when using road segments as the basic study unit. Previous literature has limited findings on the mobility data fusion and feature extraction for UFZ classification from the perspective of contextual and topological connections represented by the human movements between UFZs. Effective incorporation of the spatial relationships between the urban units using human mobility data on the UFZ level remains a research gap.

This study proposes a graph-based multimodal data-fusion framework for UFZ classification that leverages remote sensing, social sensing data, and human mobility patterns. It contributes to UFZ classification literature in three ways. First, the proposed framework investigates the efficiency of graph-based models for urban functional zone classification. Second, the contributions of latent features from multisource data are evaluated and the results demonstrate the combination of remote sensing data and social perception data that improve the accuracy of model predictions. Third, the evaluations of classification performances reveal that human mobility patterns can improve UFZ classification by mining the connections between human movements and urban functions. Additionally, the study shed light on how the classification performances are affected by multiple data sources, classifiers, and parameters of classification models through the experiments in Zhuhai and Singapore.

## 2. Study Areas and Datasets

### 2.1. Study Areas

We evaluated the proposed UFZ classification framework on two cities: Zhuhai, Guangdong Province of China, and Singapore. The locations of the two cities are presented in Figure 1.

Zhuhai (22°16′N, 113°34′E), a prefecture-level city with a population of 2.44 million in 2020, is one of China's four original special economic zones (SEZs) established in the 1980s. It covers 1724.32 km², which in addition to its mainland includes 217 islands. Zhuhai has experienced tremendous changes in urban morphological composition and landscape structure due to the rapid urbanization since it is granted the SEZ status. The changes in its urban space are attributed to various human activities, making Zhuhai an ideal location for UFZ identification. The study site focused on the mainland of Zhuhai, covering 1507.07 km², with a complex urban functional structure and a diverse range of land cover patterns. Singapore (1°29′N, 103°85′E) is a highly urbanized city-state in South-East Asia, with a population of 5.45 million. The land reclamation projects increased the

land area of Singapore to 724.2 km² by 2020. Since the past century, Singapore has been undergoing an accelerated process of urbanization with rapid land-use/land cover transformations. Our study region covers the main island area of Singapore, occupying a total area of 692.28 km², and contains various urban functional areas. The different morphological compositions and urbanization processes between the two cities provide good comparisons for UFZ classification as well as evidence to demonstrate generalizability of our proposed method.

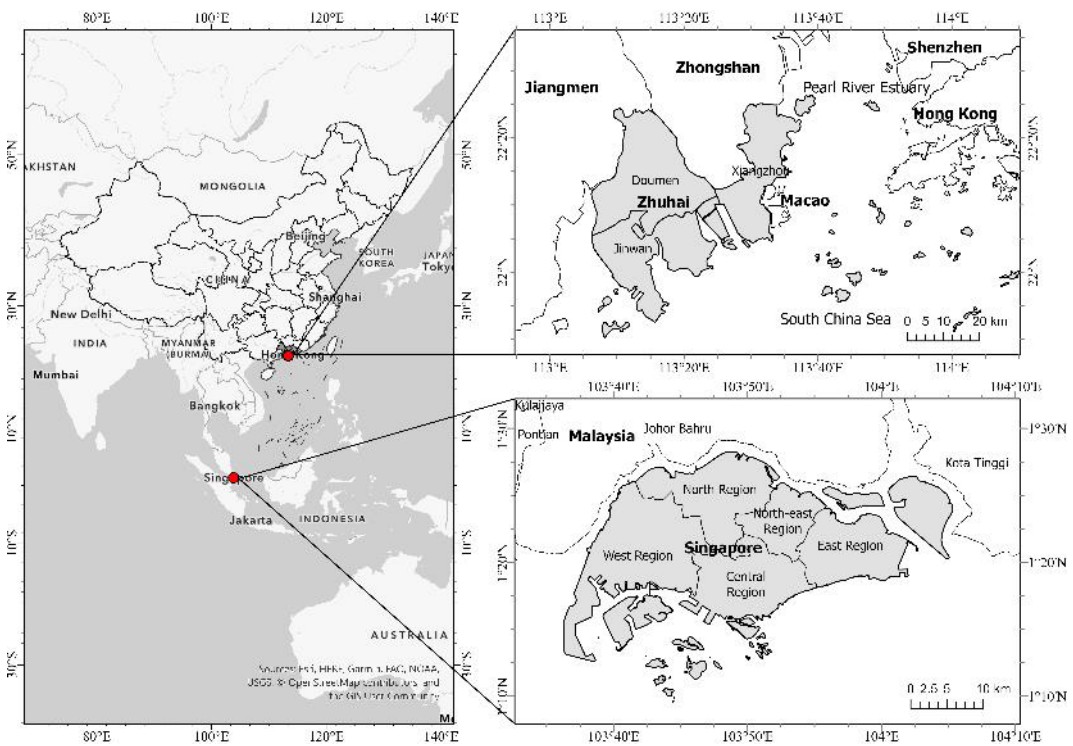

**Figure 1.** Geographic locations of study areas.

### 2.2. Datasets

The corresponding data sources, including HSR remote sensing images, POIs, building footprints, forest canopy height, OpenStreetMap (OSM) road networks, and trajectory data were mainly collected from crowdsourced datasets and government websites as described in Table 1. HSR remote sensing images with three bands (i.e., red, green, and blue) covering both of the study areas in 2019 were collected from Google Earth, and the spatial resolution is 0.6 m. These HSR images contain rich geometric, textural, and spectral information of geographic objects that enables extracting physical features of urban subregions useful for UFZ classification. The POI data of Zhuhai in 2018 were extracted from Gaode map (lbs.amap.com/. Accessed 1 Dec. 2020), in which POIs were point features with information including latitude, longitude, and triple category. The POI data of Singapore were derived from OSM, google places, and a government website (Data.gov.sg) in 2019. The 3D building data of Zhuhai were collected from the 3D maps produced by Baidu map. The original datasets include 3D building objects with attributes of height, shape, and area. The 3D building data of Singapore were collected from OSM buildings (3dbuildings.com/. Accessed 7 Dec. 2020). The forest canopy height data were extracted from a raster dataset that measured the global canopy height in 2019 produced by NASA [36]. The vector data of OSM road networks were obtained from OSM in 2019, including administrative boundaries of Zhuhai and Singapore. Figure 2 and Figure 3 show a section in Zhuhai and in Singapore as examples of data used in this study.

**Table 1.** Data sources.

|  | Data used | Time | Spatial information | Data source |
|---|---|---|---|---|
| Zhuhai | OSM data | 2019 | parcel-based | OpenStreetMap |
|  | HSR imagery | 2019 | 0.6 m/pixel | Google Earth |
|  | POI data | 2019 | point-based | Baidu map |
|  | Forest canopy height data | 2019 | 30 m/pixel | Global Forest Canopy Height [35] |
|  | Building data | 2019 | parcel-based | Baidu map |
|  | Taxi GPS data | 2019/8/01-2019/8/31 | point-based | Didi taxi dataset |
| Singapore | OSM data | 2020 | parcel-based | OpenStreetMap |
|  | HSR imagery | 2020 | 0.6 m/pixel | Google Earth |
|  | POI data | 2020 | point-based | Data.gov |
|  | Forest canopy height data | 2019 | 30 m/pixel | Global Forest Canopy Height [36] |
|  | Building data | 2020 | parcel-based | OpenStreetMap |
|  | Mobility data | 2020/9/01-2020/9/30 | point-based | CITYDATA |

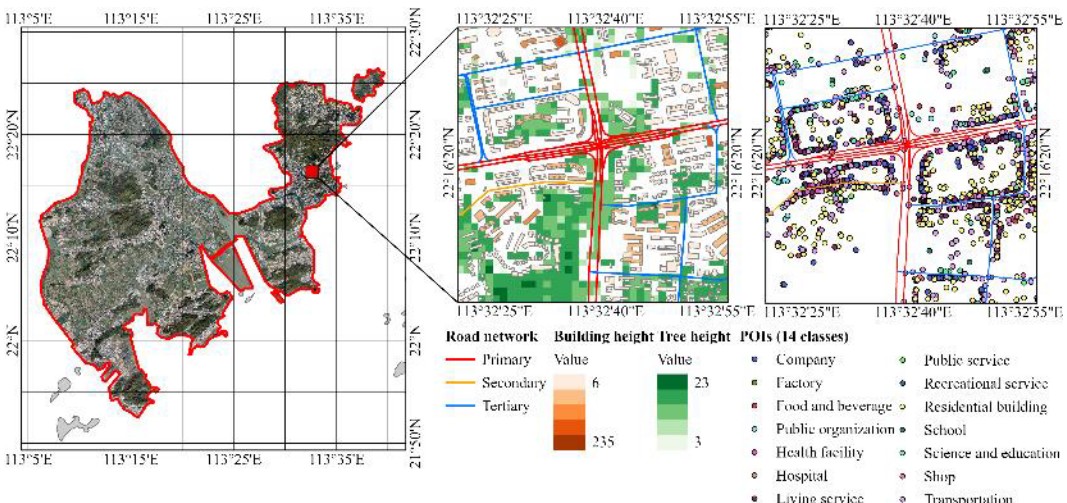

**Figure 2.** HSR image, road network, building data, forest canopy height, and POIs in Zhuhai.

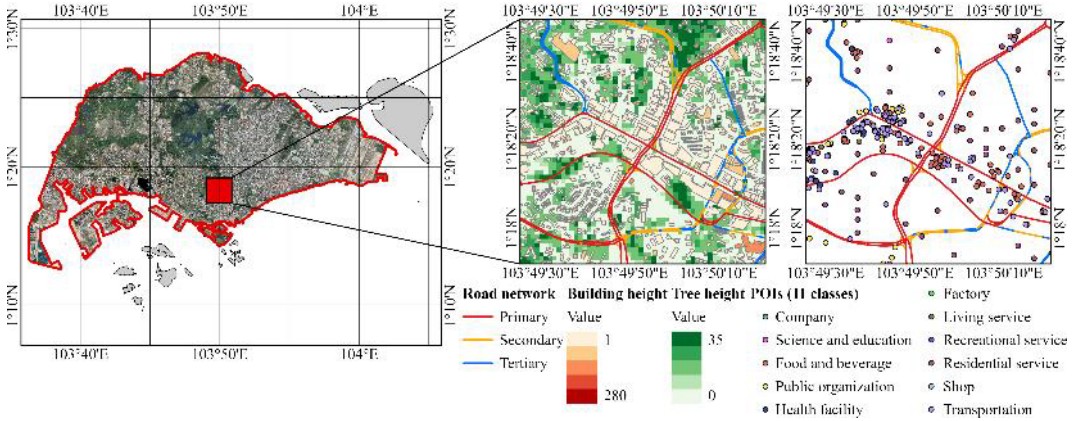

**Figure 3.** HSR image, road network, building data, forest canopy height, and POIs in Singapore.

Trajectory data, such as taxi GPS trajectory data and location-based service data, reveal the socioeconomic activities that people engage in since these activities lead to commuting between different locations [23,37], and it was suggested that taxi trips account for

a significant portion of urban residents' mobility [38]. To map the patterns of mobility across different urban zones, the GPS trajectory dataset generated by 9,075 taxis in Zhuhai between 1 August 2019 and 31 August 2019 was used to extract the pick-up and drop-off locations (i.e., O/D points) for trips. The original dataset includes information of taxi ID, longitude, latitude, timestamp, direction, speed, and the number of passengers. The crowdsourced mobility data in Singapore were obtained from CITYDATA [citydata.ai/](citydata.ai/). Accessed 1 Jun. 2021). The trajectory dataset includes 368,135 trip hops obtained from 5000 location-based service devices in September 2020.

## 3. Methods

In this paper, a graph-based analysis framework is proposed to classify UFZ by integrating multisource data. As shown in Figure 4, the proposed framework consists of four components, UFZ segmentation, multisource feature fusion, mobility graph construction, and graph-based classification. First, the spatial units for UFZ identification were segmented using OSM road networks. Then, multisource feature vectors describing the physical and socioeconomic characteristics of UFZs were developed from HSR images, POIs, building footprints, and forest canopy height maps. Next, the UFZs, along with multisource features, were converted into nodes with semantic embeddings of directed graphs based on the trajectory information extracted from mobility data. In the last step, the Graph SAGE model was introduced to identify the categories of UFZs based on the geo-semantic embeddings and mobility connections.

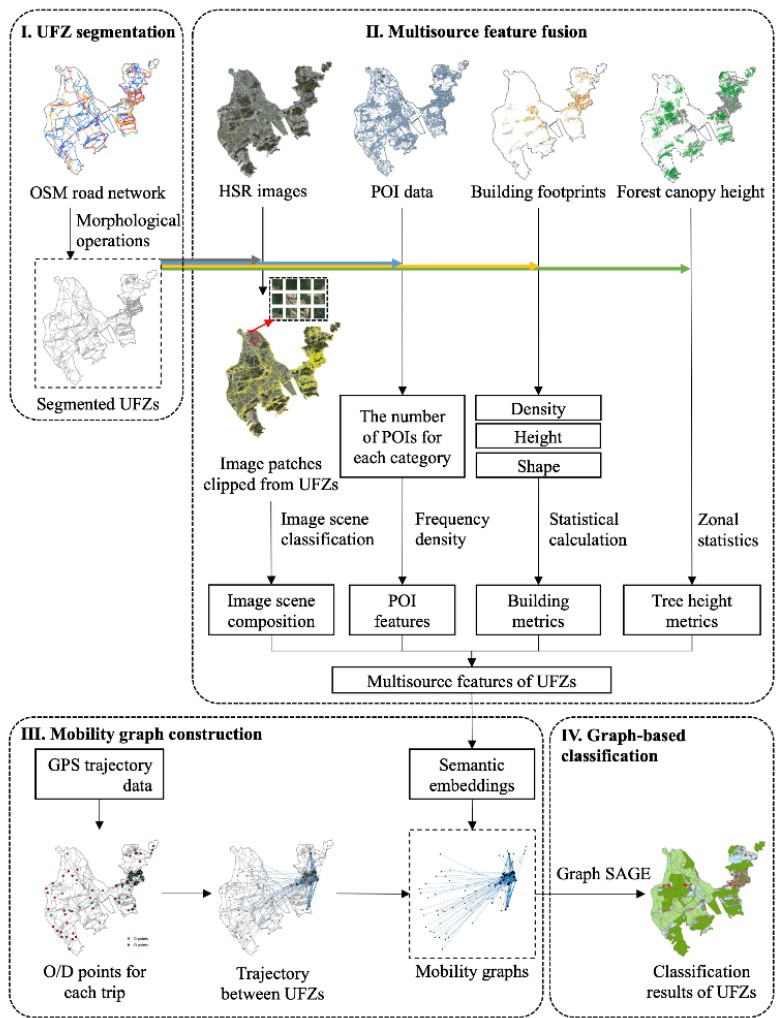

**Figure 4.** The proposed UFZ classification framework.

*3.1. UFZ Segmentation*

In urban functional identification analysis, the traffic analysis zones (TAZs) based on road network are commonly adopted as the basic research unit [37,39,40]. In this study, the TAZs segmented from multilevel road networks form the initial UFZs. UFZ segmentation was conducted using the OSM road network of two study areas dividing the study areas into multiple TAZs. We used the primary, secondary, and tertiary roads from OSM as the main road networks primarily since they offer the appropriate resolution for delineating functional zones that avoided smaller or tiny zones that do not carry meaningful activities with respect to common urban functions.

The original OSM road network in vector format was projected onto the HSR images and subsequently transformed into raster image patches with the same resolution as the HSR images. Morphological operations, including dilation and thinning, were applied to eliminate the overpasses and outlying roads in the raster road network. The binary road network in a raster format was then converted into the (vector) polygons by creating multipart features based on the connectivity between pixels. The boundaries of TAZs obtained from morphological operations do not correspond exactly to the road centerlines. The skeleton road network obtained after morphological operations divided the study areas with a buffer distance from the road centerlines, which facilitated the analysis of mobility connections between UFZs based on the origin/destination (O/D) points extracted from taxi trajectories or other mobile devices.

*3.2. Multisource Feature Extraction*

The UFZ classification differs from land use/land cover (LULC) classification since UFZs are heterogeneous zones composed of multiple objects with different LULC types [22]. As a result, the spatial distribution of ground components and their types plays significant roles in UFZ recognition. In addition to these physical attributes, socioeconomic attributes are equally essential, given that UFZs are defined with human activities which are a major consideration. To capture the physical and socioeconomic attributes, the multisource data introduced in Section 2.2 are employed, and semantic features were extracted and then integrated to characterize UFZs.

3.2.1. HSR Image Scene Composition

UFZs are composed of heterogenous ground objects, such as buildings, playgrounds, and green lands. To classify UFZs, the physical characteristics of images at the pixel level are commonly used [20,21,28,41]. Nevertheless, pixel-level features often fail to represent the socioeconomic functions of an urban zone since the UFZ is composed of various socioeconomic information bearing objects—a residential UFZ may include residential buildings, roads, and green land, and a public service UFZ may include functional buildings, playgrounds, and green land. In this study, rather than pixel-level features, we opted for describing UFZs through scene composition. Here, a scene refers to a square image patch representative of a ground object and the composition refers to the proportions of individual ground objects within the UFZ. Scene composition is adopted since the socioeconomic functions of UFZ can be inferred through the occurrences of ground objects. Specific steps to extract scene composition are described in the next section.

3.2.2. Image Scene Composition Extraction

To obtain the scene composition, a CNN-based method, which uses the hierarchical deep architecture of CNN to automatically learn high-level features of remote sensing images, is generally used. In this study, we adopted Resnet-50, a widely used CNN-based method that achieved high accuracy in the scene classification task in several benchmark remote sensing scene classification datasets [42–44].

The extraction of image scene composition features was realized by the following steps (Figure 5). First, we generated the datasets of image scenes with a size of 256×256

clipped from each UFZ. A total of 32,203 image scenes were randomly selected from Zhuhai and Singapore image datasets. Second, we labeled the image scenes with 11 classes, including agriculture, bare land, commercial buildings, forest, functional buildings, green land, industrial, playground, residential, road, and water, based on the categories from the land use maps of Zhuhai and Singapore obtained from the global land cover dataset [45]. The sample dataset was divided into a training set, a validation set, and a test set by the ratio of 6:1:3. Third, Resnet-50, which was pretrained on ImageNet and later fine-tuned with the sample data to reach 88.7% classification accuracy on our test set, was employed to predict the classes of all the image scenes generated within UFZs. The number of scenes for each land use class was counted for each UFZ, the percentages of each scene class are considered as the features of image scene composition. After classifying the HSR image patches based on deep learning features, the image scene composition of each UFZ can be represented by a vector with 11 elements, with each one reflecting the percentage of a land use category within the UFZ. For $r$-th UFZ, the feature vector of scene distribution is represented as:

$$s_r = c_{1,r}, c_{2,r}, c_{3,r}, \ldots, c_{11,r}, \ r \in 1,2,3,\ldots,n \tag{1}$$

where $c_{1,r}, c_{2,r}, c_{3,r}, \ldots$ indicates the percentage of different classes of land use appearing in $r$-th UFZ and $n$ is the total number of UFZs.

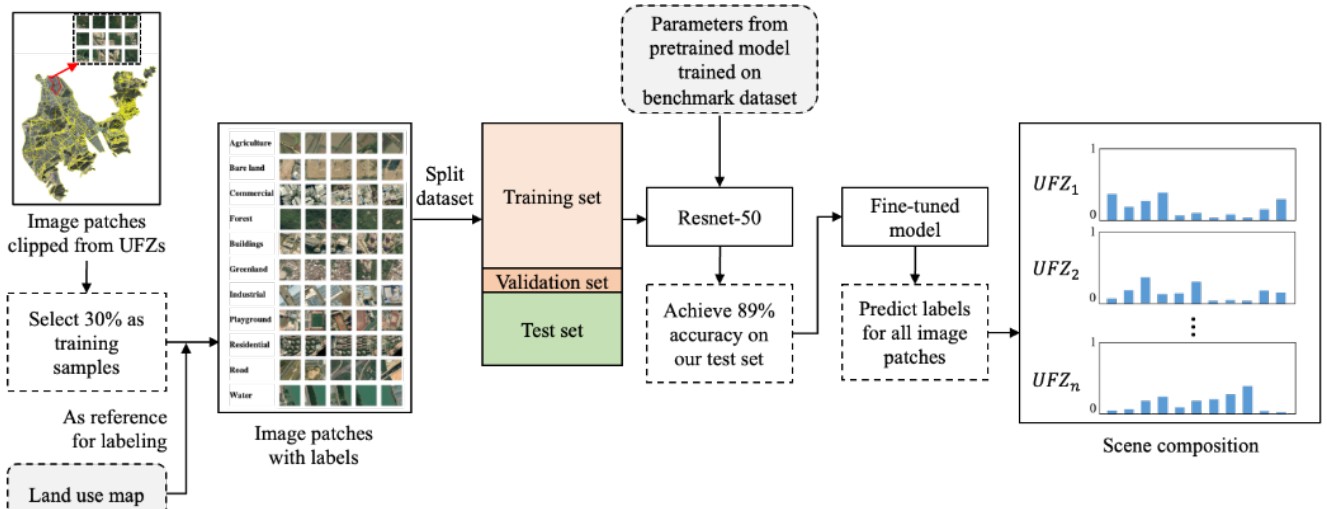

**Figure 5.** Procedures for extracting image scene compositions for UFZs.

### 3.2.3. Socioeconomic Features from POIs

POI categories can be viewed as virtual words that reflect socioeconomic properties. Therefore, the number and distribution of POIs in each UFZ indicate the land use patterns and socioeconomic functions. The original POI data collected in Zhuhai include 23 classes of POIs that are not entirely mutually disjoint, resulting in duplication and ambiguity. Therefore, we reclassified level-1 POIs into 14 categories by merging or splitting some of their classes, as displayed in Table 2. For example, places such as schools and hospitals represent specific social functions; therefore, they were divided into independent classes. Transportation service, road facility, and pass facility have similar functions, and then they were merged into the category of public services. Correspondingly, the POIs in Singapore were reclassified into 11 classes by integrating categories with similar functions, as shown in Table 3.

The socioeconomic characteristics of UFZ are described through the spatial distribution of different categories of POIs. To minimize the biases caused by unbalanced data volumes between POI categories, the POI feature vector is constructed using the frequency density of $i$-th POI category, defined as:

$$v_{i,r} = \frac{POI_{i,r}}{Area_r}, i \in \{1, 2, \ldots, \mathrm{m}\}, r \in \{1, 2, \ldots, \mathrm{n}\} \tag{2}$$

where $v_{i,r}$ is the frequency density of $i$-th POI category in $r$-th UFZ, $POI_{i,r}$ represents the number of POI points of $i$-th POI category within $r$-th UFZ, and $Area_r$ is the total area of $r$-th UFZ. While $m$ is the total number of POI categories (i.e., Zhuhai has 14 POI classes and Singapore has 11 POI classes) and $n$ is the total number of UFZs. The POI feature vector of $r$-th UFZ is denoted as follows:

$$p_r = (v_{1,r}, v_{2,r}, \ldots, v_{m,n}) \tag{3}$$

**Table 2.** Statistics of segmented UFZs and POIs of Zhuhai data.

| Zhuhai data | Attribute | Count |
|---|---|---|
| UFZs | Initial segmented UFZs | 1276 |
| POIs | **Category** | **14** |
| | Company | 14228 |
| | Factory | 379 |
| | Food and beverage | 19231 |
| | Government agencies and public organizations | 4277 |
| | Health facilities | 3375 |
| | Hospital | 101 |
| | Living services | 33122 |
| | Public services | 749 |
| | Recreational services | 2184 |
| | Residence | 17697 |
| | School | 354 |
| | Scientific institutions and educational services | 4689 |
| | Shops | 30775 |
| | Transportation facilities | 15450 |
| | **Total counts** | **146611** |

**Table 3.** Statistics of segmented UFZs and POIs of Singapore data.

| Singapore Data | Attribute | Count |
|---|---|---|
| UFZs | Initial segmented UFZs | 886 |
| POIs | **Category** | **11** |
| | Company | 1599 |
| | Industry | 607 |
| | Food and beverage | 1599 |
| | Government agencies and public organizations | 1759 |
| | Health facilities | 1600 |
| | Living services | 1597 |
| | Recreational services | 2032 |
| | Residence | 3206 |
| | Scientific institutions and educational services | 638 |
| | Shops | 1600 |
| | Transportation facilities | 2913 |
| | **Total counts** | **19150** |

### 3.2.4. Morphological Features of Buildings and Trees

The building features are calculated based on the building footprints with 2D and 3D information. The metrics used in our studies are presented in Table 4. For building metrics, nine indicators are used to describe the structures of buildings in UFZ parcels based on their physical properties, including three 2D metrics—area, perimeter, and building

structure ratio—and one 3D metric—height. Building structure ratio is calculated by dividing the building area by the perimeter. These metrics are included as they tend to exhibit different levels of variations, e.g., commercial UFZs tend to exhibit high variations whereas residential UFZs tend to exhibit less variations. In addition, commercial UFZs tend to have more complex building shapes. Building heights, which reflect 3D characteristics of the buildings, can potentially contribute to UFZ classification since they may reflect the different activities that take place within, e.g., commercial buildings tend to be higher while residential buildings tend to be of similar heights. Along with other metrics considered in this study, they account for important urban morphological attributes indicative of varying activity intensities.

In addition to the 2D and 3D metrics of buildings, tree canopy height is included as a metric in this study as its variations reflect the morphology of urban green space, which helps differentiate green spaces and agricultural UFZs from others. Accordingly, the mean value and standard deviation (std) value of tree canopy height are regarded as significant indicators for UFZ classification. The feature vector of 2D and 3D metrics for $r$-th UFZ is shown as:

$$m_r = \{ba\_ensity_r, ba\_mean_r, ba\_std_r, be\_mean_r, be\_std_r, bsr\_mean_r, bsr\_std_r,$$

$$bh_mean_r, bh_std_r, th_mean_r, th_std_r\}, r \in \{1, 2, \ldots, n\} \tag{4}$$

### 3.2.5. Feature Generation

The feature vectors described from Sections 3.2.1 to 3.2.3 form the complete set of features for distinguishing individual UFZs. In this study, these feature vectors were concatenated in order that the full set of semantic features of each UFZ can be readily used in the proposed graph-based approach. The concatenated semantic feature vector for $r$-th UFZ is represented as:

$$x_r = \{s_r, p_r, m_r\}, r \in \{1, 2, \ldots, n\} \tag{5}$$

**Table 4.** Morphological indices of buildings and tree height.

| | Morphological Index | | Description |
|---|---|---|---|
| 2D metrics | Building area | ba_density | Building density in one UFZ |
| | | ba_mean | Mean of building area |
| | | ba_std | Standard deviation of building area |
| | Building perimeter | be_mean | Mean of building perimeter |
| | | be_std | Standard deviation of building perimeter |
| | Building structural ratio | bsr_mean | Mean of building structure ratio |
| | | bsr_std | Standard deviation of building structure ratio |
| 3D metrics | Building height | bh_mean | Mean of building height |
| | | bh_std | Standard deviation of building height |
| | Tree height | th_mean | Mean of tree height |
| | | th_std | Standard deviation of tree height |

### 3.3. Graph-based UFZ Classification

To utilize trajectory information, it is required to first construct the corresponding mobility graph before the second, which is to classify individual zones using a combination of the features mentioned in the preceding section and the constructed graph.

### 3.3.1. Construction of Mobility Graph

Graph theory is widely used in analyzing the spatial distributions of remote sensing image units and modeling GIS data-based transportation networks. Specifically, studies in human mobilities have shown that trajectories, which lead naturally to graph-based

models, are one of the indicators for human activities associated with the socioeconomic characteristics of UFZs [26,37,46]. Transforming UFZs to mobility graphs is a critical step in the graph-based classification framework. Following the calculations of multisource features for initial UFZs, we integrated the trajectory information with topological linkages of UFZs to capture the contextual information for classifying UFZs.

**Table 5.** Statistics of trajectory data used in our study.

| Data Source | Attributes | Count |
|---|---|---|
| Zhuhai taxi GPS dataset | Taxis | 4390 |
| | Effective days | 30 |
| | Pick-up points | 44654 |
| | Drop-off points | 42718 |
| | Trajectories | 17955 |
| Singapore mobility dataset | Mobile devices | 4738 |
| | Effective days | 30 |
| | Leaving points | 368135 |
| | Arriving points | 368135 |
| | Trajectories | 21647 |

The preprocessing of trajectory data includes extracting O/D points, cleaning outlier data points, and recording effective trips between corresponding origin and destination points. The statistics of preprocessed trajectory data are shown in Table 5**.** First, we locate O/D points in UFZs. The directions of trajectories from origin points to destination points represent the human movements between UFZs. Next, the UFZs are treated as nodes in a graph with multisource features and the directions of trajectories between UFZs represent the connections between UFZ nodes. The UFZ nodes are connected based on adjacency and directions of trajectories to form a mobility network from Figure 6a to Figure 6b. Finally, we use a mobility network to construct a directed graph $G = (V, E)$, consisting of a set of nodes $V$ that are connected by edges $E$. Nodes $V$ refer to UFZs and edges $E$ refer to mobility connections between UFZs. Each UFZ node in the mobility graph is deemed as a node $v_i$ and each trajectory as an edge $e_{i,j}$, with the direction of edge from the start node $v_i$ to the end node $v_j$ of each trajectory. Each node $v$ has a feature vector $x_v$ , which is summarized in a vector matrix $X \in R^{n \times d}$, where $n$ represents the number of nodes (i.e., the number of UFZs) and $d$ is the dimension of the feature vector (i.e., the number of features), each row $x_v \in R^d$ is the semantic feature vector for *node v*, as shown in Figure 6c. The adjacency matrix of *graph G* is defined as $A \in R^{n \times n}$.

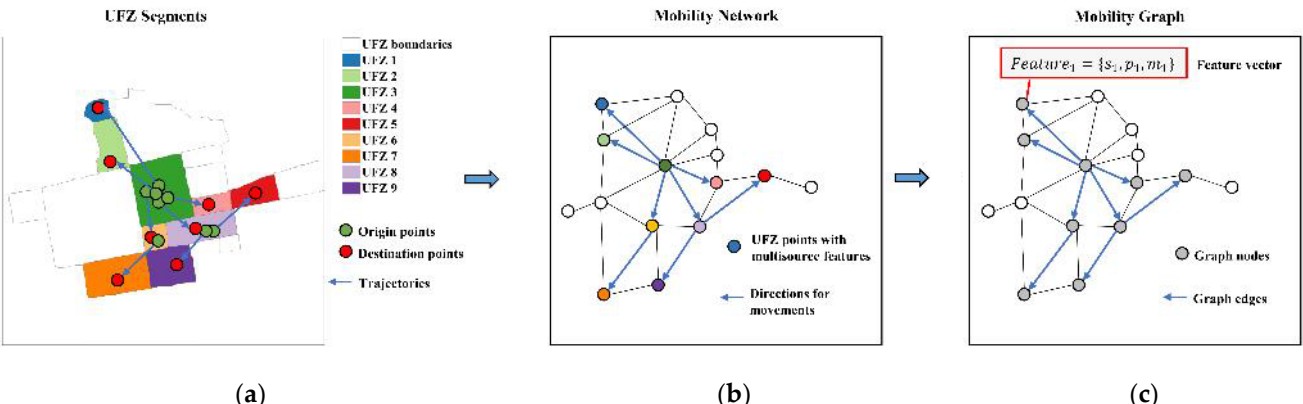

(**a**)          (**b**)          (**c**)

**Figure 6.** UFZ mobility graph generation: (**a**) UFZs contain O/D points and effective trajectories; (**b**) mobility network constructed based on UFZ points and movements; (**c**) mobility graphs with multisource feature embeddings transformed from mobility network.

It should be noted that, while the *graph G* includes all UFZs as nodes, it is entirely possible that some nodes do not serve as an origin or a destination, i.e., no one boarded

or alighted on these UFZs. In Figure 6c, these UFZs were presented by the hollow nodes. For these nodes, their labels or types will not participate in the Graph SAGE classification process described in the next section; rather, their classification will be based only on non-mobility features described in Section 3.2.

### 3.3.2. Classification of UFZ using Graph SAGE

To take advantage of the topological connections and the human movement information built in the graph for UFZ classification, this study considered a graph convolution approach to aggregate the spatial context information of each UFZ. The approach contains two convolutional layers and a *softmax* layer that was used as the final classifier. The Adam algorithm was adopted to optimize the loss and predicted labels for nodes in the graphs. The algorithm used back propagation according to the difference between the predicted label and the ground truth label, and optimized the network weight until the weight parameter with the best prediction result was obtained. The training set and test set split randomly from the input graph at first and the attribute of each node is with the same dimensions as the feature vector. Spatial context information of each node was first extracted through two layers of graph convolutional layers. The dimension of the extracted feature was further reduced to six through a fully connected layer to match the number of categories. Then, the model prediction was obtained by applying *softmax* on top of it and selecting the category with the largest probability.

In this study, the UFZs are regarded as graph entities and trajectories are transformed into directed edges. Following the construction of a directed graph based on the mobility features, two graph structures, graph convolutional networks (GCN) and Graph SAGE (SAmple and aggreGatE), were employed to identify the categories of nodes or UFZ categories.

GCN, a two-layer model, operates directly on a graph and induces embedding vectors of nodes based on the properties of their adjacent neighbors [47]. In the graph-based model, the graph semantic feature matrix $X$ and the adjacency matrix $A$ are taken as inputs for the multilayer GCN computation. The output for the first layer of GCN is represented as:

$$H^{(1)} = \sigma(\tilde{A}XW_0), \tag{6}$$

where $W_0 \in R^{m \times n}$ is a weight matrix of trainable parameters at the first layer and $\sigma$ is an activation function, e.g., a rectified linear unit (ReLU) $\sigma(x) = max(0, x)$. We can aggregate high order neighborhood information by stacking multiple GCN layers:

$$H^{(l+1)} = \sigma(\tilde{A}L^{(l)}W_l), \tag{7}$$

where $l$ denotes the layer number and $H^{(0)} = X$. When multiple GCN layers are stacked together, information about larger neighborhoods is captured, which reveals the spatial dependencies among UFZs. However, the simple GCN layers cannot reveal the hidden connections between nodes. To deeply discover the relationships between nodes and edges in a graph, we introduced Graph SAGE for classifying UFZs based on constructed mobility graphs. Developed from the structure of simple GCN layers, Graph SAGE is a general inductive framework that efficiently generates node embeddings for previously unseen data by leveraging node feature information (e.g., text attributes) [48]. The aggregation process of Graph SAGE is illustrated in Figure 7.

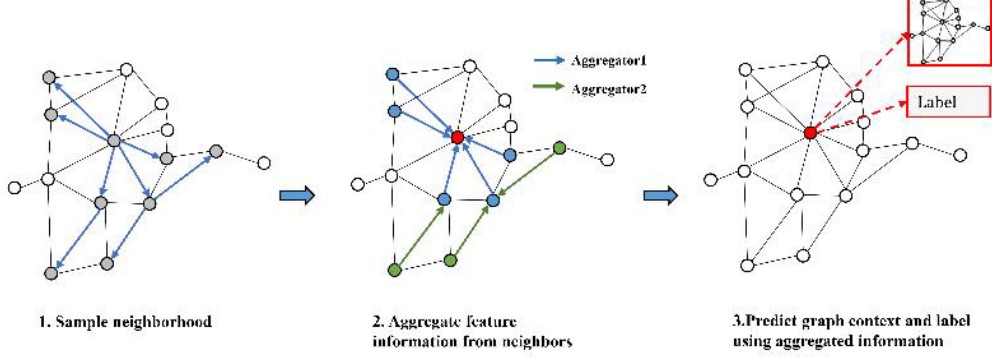

**Figure 7.** Graph SAGE aggregation process.

Rather than training a unique embedding vector for each node, a set of aggregator functions is trained that learns to aggregate feature information from a node's immediate neighborhood. The data away from a given node are aggregated by each aggregator function from a set of search depths. Graph SAGE generates embeddings for previously unseen nodes by applying learned aggregation functions at test or inference time. The forward propagation rule for Graph SAGE is expressed as Algorithm 1.

---

**Algorithm 1:** Graph SAGE embedding generation (i.e., forward propagation) algorithm

---

**Input:** Mobility graph $G = (V, E)$ constructed based on O/D points and UFZs; multi-source features: $x = \{x_1, x_2, \ldots, x_v, \forall v \in V\}$; the number of layers of the network $K$; non-linearity $\sigma$; mean aggregator functions $AGG$; neighborhood function $N: v \rightarrow 2^v$

**Initialization:**

1: $h_0 \leftarrow x_v, \forall v \in V$

2: for k = 1 to K do

3:     for $v \in V$ do

4:         $h_{N(v)}^k \leftarrow AGG(\{h_v^{k-1}, \forall u \in N(v)\})$

5:         $h_v^k \leftarrow \sigma(W^k \cdot CONCAT(h_v^{k-1}, h_{N(v)}^k))$

6:     end

7:     $h_v^k \leftarrow h_v^k / \|h_v^k\|_2, \forall v \in V$

8:     $z_v \leftarrow h_v^k, \forall v \in V$

9: end

**Output:** Vector representations for all $v \in V$

---

From Algorithm 1, $K$ is the number of layers of the network, $\forall u$ is the eigenvector of the node $u$, $\{h_v^{k-1}, \forall u \in N(v)\}$ denotes the embeddings of the neighbor $u$ of the node $V$ in the $k - 1$ layer, and $h_v^k$ represents the characteristics of all neighbors of node v at the k level. For each iteration, the nodes collect information from their local neighbors, and as the process continues, the nodes acquire progressively more information from the farther reaches of the graph. Therefore, the extraction of long-range contextual linkages occurs. The output of multilayer Graph SAGE is $H^K$. The cross-entropy error is utilized to punish the disparity between the network output and the labels of the original labeled samples, specifically:

$$\mathcal{L}_{classification} = -\frac{1}{c} \sum_{i=1}^{C} y_i * \log(softmax(FC(H^K))) \tag{8}$$

where $y_i$ is the labeled examples set, C denotes the number of classes, and $H^K$ is the output of Graph SAGE with K layers. The detailed procedures of Graph SAGE classification are

shown in Algorithm 2. We considered feature matrix $X$ as the inputs and trained the multilayer network using the UFZ mobility graph $G$ as in Algorithm 2. We used two Graph SAGE convolutional layers to obtain the hidden relationships between UFZs. The number of epochs is set to 1000, while the learning rate is set to 0.001 and dropout to 0.2. The Graph SAGE model was constructed and trained through iterations. The output was the predicted label vectors for UFZs. In this study, we split the origin datasets by 7:3 for training and testing, respectively. We examined the impacts of different feature combinations on functional zone classification by combining different features as the inputs for various classification models.

---

**Algorithm 2:** Proposed graph-based framework for UFZ classification

**Input:** Multisource features of UFZs; trajectories between UFZs; number of graph convolution layers = 2; number of epoch T; learning rate = 0.001; dropout=0.2; Adam gradient descent; python =3.7; pyTorch = 1.7.1

1: Extract edge list from trajectories and node embeddings from multisource features;
2: Construct mobility graph $G = (V, E)$
3://Train Graph SAGE model
4: for t = 1 to T do
5:　　//Graph convolution nodes feature
6:　　Perform graph learning at adjacent points spatial level by Algorithm 1
7:　　Batch normalization, dropout, and ReLU
8:　　Perform graph learning at adjacent points and farther points spatial level by Algorithm 1
9:　　Batch normalization, dropout, and ReLU
10:　　Output the graph leaning feature of all nodes
11:　　Calculate the error term according to Equation (8) and update the weight matrices using Adam gradient descent
12: **end for**
14: Conduct label prediction for all nodes based on the trained model
**Output:** Predicted label for each UFZ

---

## 4. Results

*4.1. Classification Results Using Graph-Based Models*

### 4.1.1. Results of the Zhuhai Dataset

The proposed graph-based classification framework combines urban scenes, building objects, canopy height, and socioeconomic attributes to construct semantic features of functional zones. The spatial heterogeneity of ground objects and the variations in urban morphology reflect the distinct social functions of each urban subregion in different cities. We defined six types of functional zones in our study: Agriculture zones, commercial zones, industrial zones, public service zones, green space zones, and residential zones, which were set by referencing existing UFZ classification systems and published papers [7,8,11,13,21]. The classified UFZ map in Zhuhai is presented in Figure 8 (a).

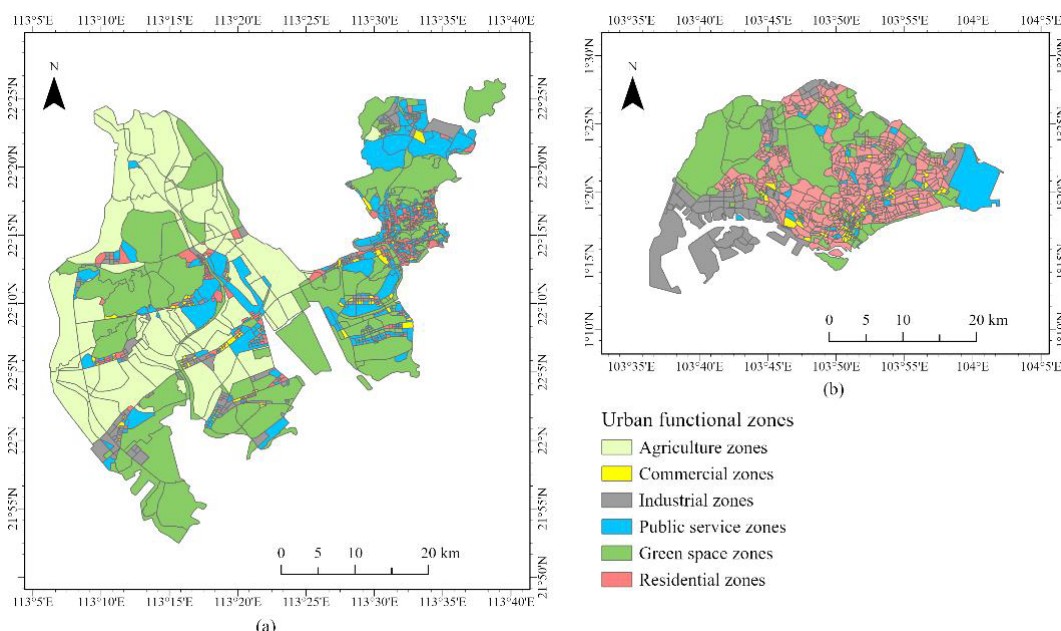

**Figure 8.** UFZ classification maps based on Graph SAGE model: (**a**) The UFZ map of Zhuhai; (**b**) the UFZ map of Singapore.

The proportion of scene distribution of each UFZ is extracted based on the deep learning features of HSR remote sensing imageries. The socioeconomic characteristics are described through the spatial joining between POI densities and urban zones. Aside from the 2D physical features of buildings, the 3D morphological indices are calculated based on building objects and tree canopies. The integrated multisource features are applied to the graph-based model for UFZ classification in Zhuhai. The confusion matrix obtained by the proposed method using multisource features is shown in Table 6. The overall classification accuracy of our method is around 75.79% when using combined features on Zhuhai dataset. The commercial zones and public service zones have relatively lower producer's accuracy compared to other UFZ categories. Some public service zones in the test set are mistakenly classified into residential and commercial zones. The public service zones where administrative agencies and scientific institutions are located may include residential buildings and POIs for food and beverage or living services, which makes this type of UFZ difficult to be distinguished from others.

Figure 9 zooms into the UFZs in Zhuhai which are misclassified by our proposed method. Most misclassifications are found in the central urban areas in the eastern part of Zhuhai. In correspondence to the confusion matrix, the model is not good at recognizing between public service zone and other UFZs; however, it performs well in most residential and agricultural zones.

**Table 6.** Confusion matrix for the test dataset in Zhuhai using Graph SAGE model.

| Predicted<br>Actual | A | C | I | P | G | R | Producer's accuracy |
|---|---|---|---|---|---|---|---|
| A | 25 | 0 | 0 | 1 | 1 | 1 | 89.29% |
| C | 0 | 29 | 1 | 9 | 0 | 6 | 64.44% |
| I | 0 | 0 | 53 | 7 | 2 | 2 | 82.81% |
| P | 0 | 4 | 5 | 57 | 7 | 17 | 63.33% |
| G | 1 | 0 | 3 | 10 | 45 | 0 | 76.27% |
| R | 0 | 2 | 1 | 11 | 1 | 79 | 84.04% |
| User's accuracy | 96.15% | 82.86% | 84.13% | 60.00% | 80.36% | 75.24% | OA=75.79% |

A: Agriculture zone; C: Commercial zone; I: Industrial zone; P: Public service zone; G: Green space zone; R: Residential zone; and OA: Overall accuracy.

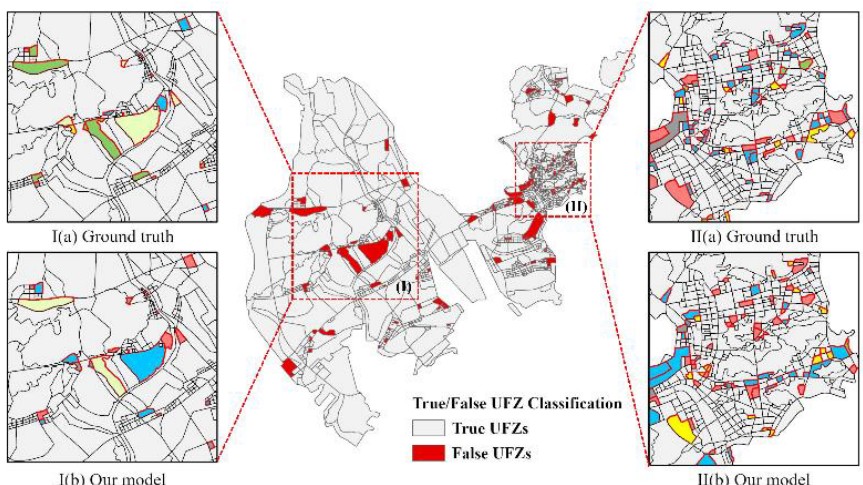

**Figure 9.** Misclassification map in Zhuhai using Graph SAGE model.

4.1.2. Results of the Singapore Dataset

We evaluated the generalizability of the proposed method using the Singapore dataset. For Singapore, 886 UFZs with ground truth labels were obtained using the semi-supervised sampling method with the land use maps in 2019. The image, building, tree, and POI features are extracted to construct the vector of semantic features. The location-based service data are transformed into trajectories to establish the mobility graph. Results of the UFZ classification map are presented in Figure 8b, and quantitative evaluations are presented in Table 7. The overall accuracy of Graph SAGE reaches 84.5%, with a producer's accuracy of about 58% for commercial zones, 93% for industrial zones, 89% for public service zones, 61% for green space zones, and 92% for residential zones.

The precisely classified numbers in the confusion matrix and evaluation metrics in the Singapore dataset demonstrated that the proposed model is suited for distinguishing the industrial and residential zones in Singapore; however, it has lower efficiency in identifying commercial and green space zones (Table 7). Figure 9 shows the mistakenly classified UFZs of our proposed method in Singapore. According to Figure 10, several commercial, public service, and residential zones were misclassified using the graph-based model. The mixed functions in one building or a community are the main reason for the misclassification. In Singapore, many shopping malls are located close to or within residential zones. Moreover, the high coverage of greenness in this city results in the complex mixture of green space with residential or public service regions, making it challenging for the model to accurately distinguish between these zones based on semantic features sharing similar patterns. Nevertheless, the experiments in Singapore demonstrate that the proposed method is applicable for UFZ identification based on mobility data.

**Table 7.** Confusion matrix for test dataset in Singapore using Graph SAGE model.

| Predicted / Actual | C | I | P | G | R | Producer's accuracy |
|---|---|---|---|---|---|---|
| C | 21 | 3 | 6 | 1 | 5 | 58.33% |
| I | 0 | 37 | 0 | 1 | 2 | 92.50% |
| P | 3 | 0 | 25 | 0 | 0 | 89.29% |
| G | 3 | 1 | 1 | 14 | 4 | 60.87% |
| R | 3 | 5 | 1 | 2 | 127 | 92.03% |
| User's accuracy | 70.00% | 80.43% | 75.76% | 77.78% | 92.03% | OA=84.53% |

C: Commercial zone; I: Industrial zone; P: Public service zone; G: Green space zone; R: Residential zone; and OA: Overall accuracy.

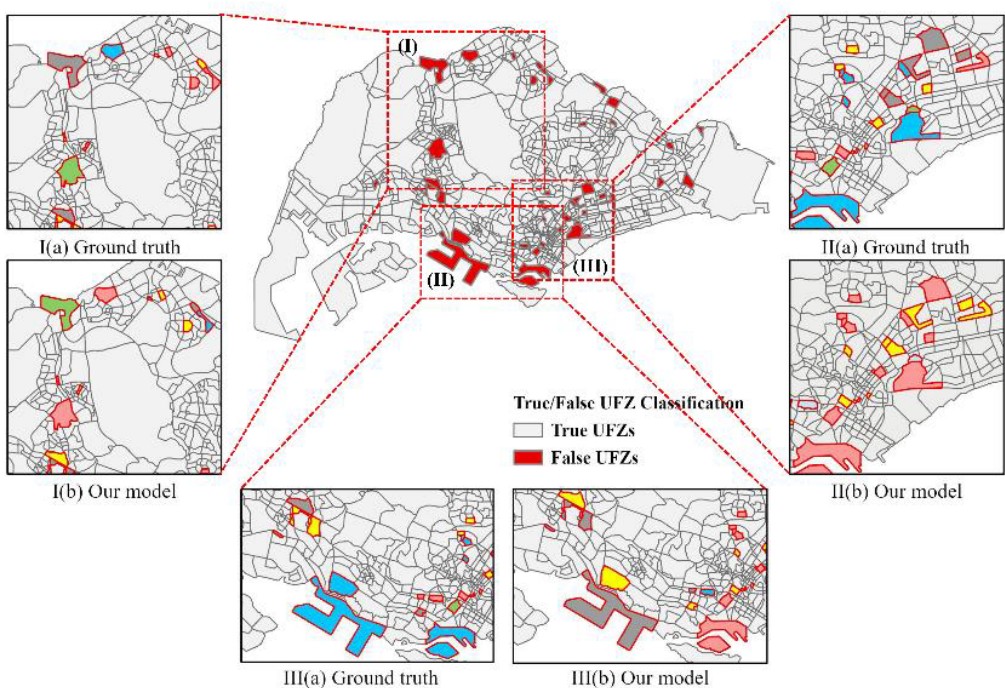

**Figure 10.** Misclassification map in Singapore using Graph SAGE model.

*4.2. Mobility patterns between Different UFZs*

Figure 11 illustrates a sample of human trajectories in Zhuhai and Singapore. For Zhuhai, we selected three taxis with the highest frequency of movements during the 30 days and plotted their movements between UFZs. The directions of movements mostly point to the central urban areas that contain more O/D points. Taxis 2 and 3 seemed to drive across the city, while Taxi 1 mainly moved around the eastern part of the city. For Singapore, we selected top seven devices with the highest frequency of movements. Figure 12 shows that the mobility of mobile device users is more consistent spatially as the O/D points and directions of trajectory are concentrated on two or three specific UFZs. The difference between the spatial patterns of trajectories of Zhuhai and of Singapore is due to the nature of the datasets we collected from the two cities; a taxi driver more commonly drives a long distance and travels to more diverse locations in a city than a white-collar or a student with mobile devices.

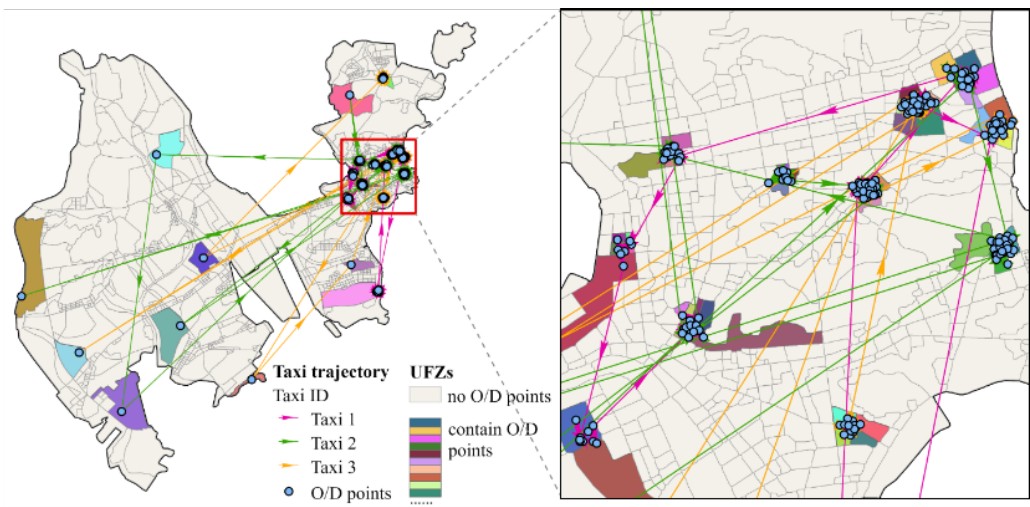

**Figure 11.** Sampled taxi trajectories in Zhuhai.

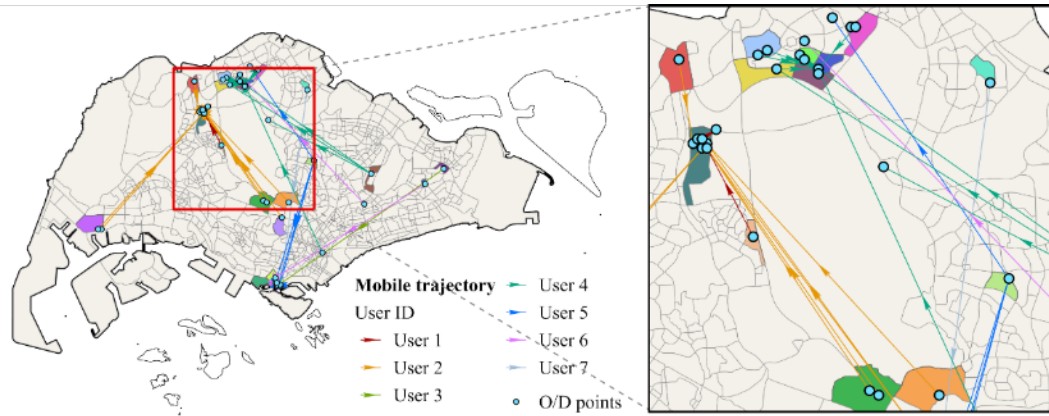

**Figure 12.** Sampled mobile trajectories in Singapore.

Based on the identified UFZ categories, we analyzed the mobility patterns within different UFZs. Figure 13 shows the flow of taxis from an origin UFZ to a destination UFZ. The movements between public service, commercial, and residential zones account for over 80% of total trajectories in Zhuhai among which the public service zone contributes to most taxi trajectories. In the Singapore trajectory dataset, the movements between residential zones account for a considerable proportion of total trajectories (Figure 14). The flow between the different UFZs indicates that after leaving a residential zone, people are more likely to travel to another residential zone, and the second highest possibility is to a green space zone. For most people leaving commercial zones, public service zones, green space zones or industrial zones, the highest possibility is to travel back to residential zones.

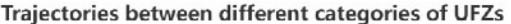

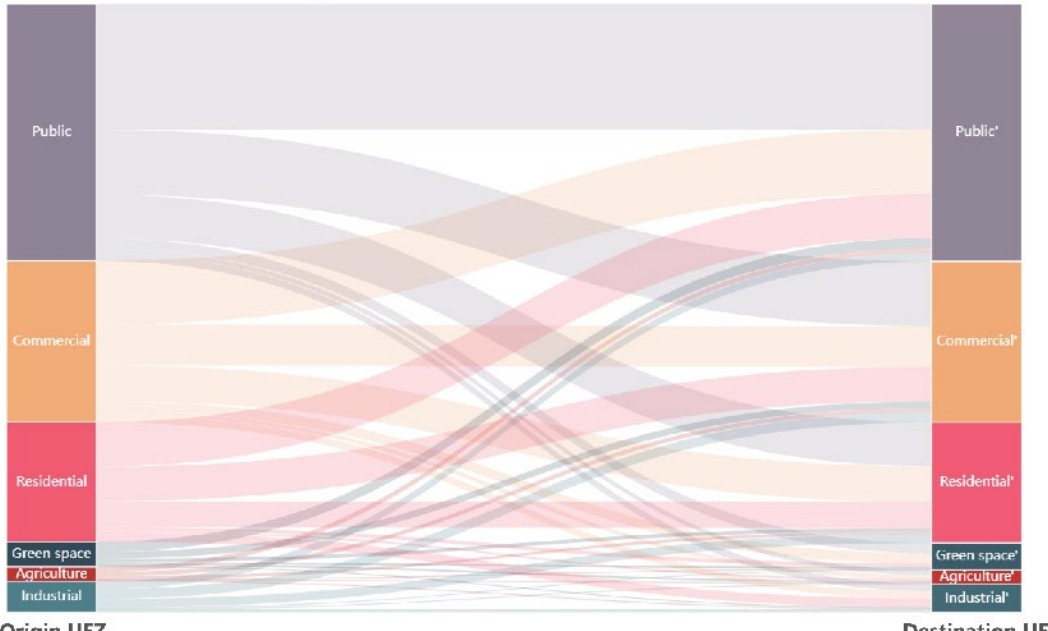

**Figure 13.** Sankey diagram of mobility flow between UFZs in Zhuhai.

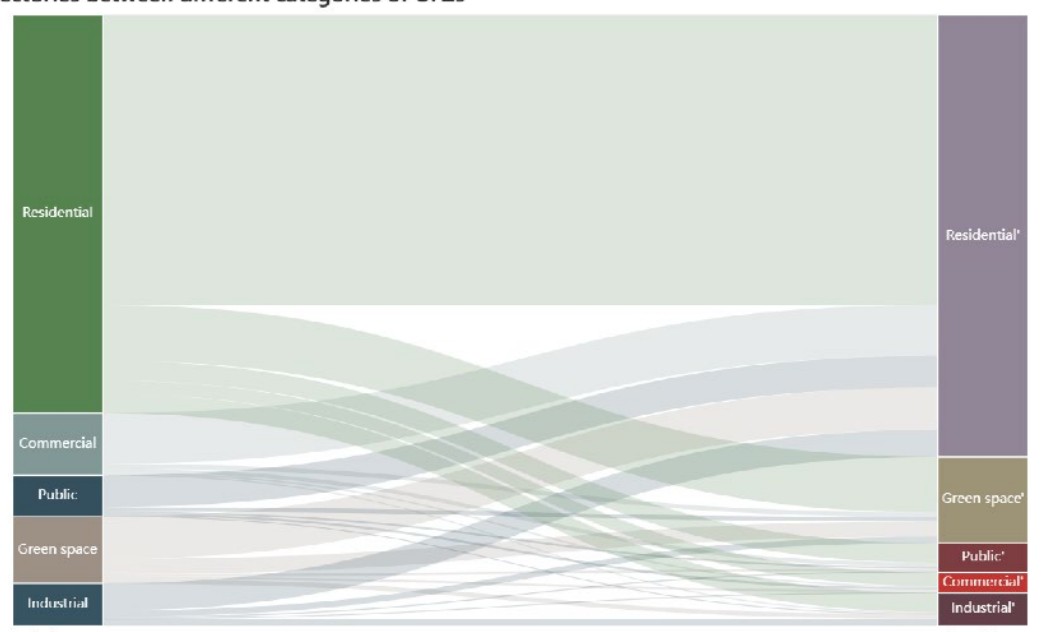

**Figure 14.** Sankey diagram of mobility flow between UFZs in Singapore.

## 5. Discussion

This section first discusses the performance of the proposed graph-based framework with different combinations of features, and the contributions of features on the UFZ classification are thus evaluated. Second, the proposed framework is compared with existing methods to validate its effectiveness. Third, the limitations of the proposed framework are discussed.

### 5.1. Comparisons of Different Feature Combinations

Tables 8 and 9 show the classification accuracies produced by various combinations of multisource semantic features in Zhuhai and Singapore. Overall, the accuracy improves as more features are considered, notably for public service zones and residential zones. When only POI data were adopted in the classification, the accuracy of agricultural zones, commercial zones, and public service zones was relatively low, which can be attributed to the low densities of POIs in these zones. For other categories, it can be observed that POI data contribute significantly to identifying UFZs. However, the incorporation of social sensing data requires caution since the data tend to be biased toward specific categories which are often clustered in specific zones. For example, in Zhuhai, the number of living service POIs is approximately forty-four times the number of public service POIs, which causes great imbalances between different UFZs.

In terms of 2D/3D morphological indices, it is observed that the integration of 2D/3D building information improves the classification accuracy in commercial zones while adding tree canopy 3D metrics contributes positively to the classification accuracy of industrial zones. The improvements can be attributed to the variations in 3D morphological landscapes of different UFZs.

In terms of physical features from HSR images, by comparing the first four rows and the last five rows in Tables 8 and 9, it can be seen that incorporating HSR features significantly increases the accuracies of UFZ classification. The results indicate that deep-learned features extracted from HSR images contribute to the identification of UFZs with heterogeneous urban scene information.

The proposed graph-based framework achieved an OA of 75.8% in Zhuhai and an OA of 84.5% in Singapore with all metrics, which demonstrated that human movements between different urban zones contributed to distinguishing spatial patterns of physical landscapes within UFZs.

**Table 8.** UFZ classification results in Zhuhai based on different feature combinations.

| Study area | Feature combination | A | C | I | P | G | R | OA | Kappa | F1-score |
|---|---|---|---|---|---|---|---|---|---|---|
| | POI | 0.18 | 0.22 | 0.72 | 0.41 | 0.69 | 0.67 | 53.1% | 0.413 | 0.513 |
| | POI + Tree | 0.92 | 0.64 | 0.81 | 0.57 | 0.70 | 0.76 | 71.7% | 0.651 | 0.717 |
| | POI + Building | 0.94 | 0.64 | 0.80 | 0.56 | 0.71 | 0.74 | 71.3% | 0.645 | 0.701 |
| | POI + Building + Tree | 0.82 | 0.64 | 0.91 | 0.54 | 0.76 | 0.80 | 73.7% | 0.674 | 0.734 |
| Zhuhai | Image | 0.89 | 0.71 | 0.88 | 0.38 | 0.54 | 0.83 | 67.6% | 0.601 | 0.666 |
| | Image + POI | 0.96 | 0.58 | 0.86 | 0.52 | 0.73 | 0.84 | 72.9% | 0.665 | 0.724 |
| | Image + POI + Tree | 0.86 | 0.71 | 0.83 | 0.58 | 0.78 | 0.79 | 73.9% | 0.677 | 0.741 |
| | Image + POI + Building | 0.96 | 0.76 | 0.77 | 0.62 | 0.71 | 0.78 | 73.9% | 0.678 | 0.744 |
| | Image + POI + Building + Tree | 0.89 | 0.64 | 0.83 | 0.63 | 0.76 | 0.84 | 75.8% | 0.722 | 0.776 |

**Table 9.** UFZ classification results in Singapore based on different feature combinations.

| Study area | Feature combination | C | I | P | G | R | OA | Kappa | F1-score |
|---|---|---|---|---|---|---|---|---|---|
| | POI | 0.72 | 0.42 | 0.61 | 0.25 | 0.94 | 74.4% | 0.605 | 0.732 |
| | POI + Tree | 0.61 | 0.65 | 0.64 | 0.15 | 0.92 | 74.8% | 0.568 | 0.728 |
| | POI + Building | 0.72 | 0.47 | 0.71 | 0.30 | 0.97 | 78.2% | 0.656 | 0.765 |
| | POI + Building + Tree | 0.75 | 0.68 | 0.82 | 0.15 | 0.98 | 81.1% | 0.715 | 0.797 |
| Singapore | Image | 0.33 | 0.85 | 0.61 | 0.40 | 0.85 | 71.8% | 0.566 | 0.710 |
| | Image + POI | 0.61 | 0.72 | 0.71 | 0.65 | 0.96 | 82.4% | 0.733 | 0.825 |
| | Image + POI + Tree | 0.58 | 0.78 | 0.82 | 0.55 | 0.93 | 82.1% | 0.728 | 0.818 |
| | Image + POI + Building | 0.69 | 0.75 | 0.82 | 0.60 | 0.91 | 82.4% | 0.734 | 0.825 |
| | Image + POI + Building + Tree | 0.58 | 0.93 | 0.89 | 0.61 | 0.92 | 84.5% | 0.763 | 0.843 |

### 5.2. Comparisions with Existing Methods

As shown in Table 10, we compared our graph-based models with traditional machine learning methods, which only utilized multisource semantic features but without mobility data for the classification. All experiments were conducted with Python language. Among all the presented traditional methods, Random Forest achieves the highest OA (73.1%) in Zhuhai, which is 2.7% lower than the Graph SAGE. According to Figure 15, the comparisons between classification accuracies of different models show that the proposed graph-based classification framework enhanced the performance of UFZ classification, and the higher OA testify that the information from human mobility data in the graph-based model contributes to identifying urban functions. The Graph SAGE model outperforms traditional classification models of random forest (RF), support vector machine (SVM), gradient boosting decision tree (GBDT), and feedforward neural network (FNN). In addition, comparisons are made between Graph SAGE and simple GCN, and it is found that OA by the Graph SAGE is 12% higher than GCN, which implies that the mobility graphs provide useful information for distinguishing between UFZ parcels when the inductive learning was integrated into the convolutional networks.

**Table 10.** Overall accuracy of UFZ classification results using different models.

| Study Area \ Model | RF | GBDT | SVM | FNN | GCN | Graph SAGE |
|---|---|---|---|---|---|---|
| Zhuhai | 0.731 | 0.722 | 0.683 | 0.644 | 0.657 | 0.758 |
| Singapore | 0.811 | 0.812 | 0.78 | 0.748 | 0.726 | 0.845 |

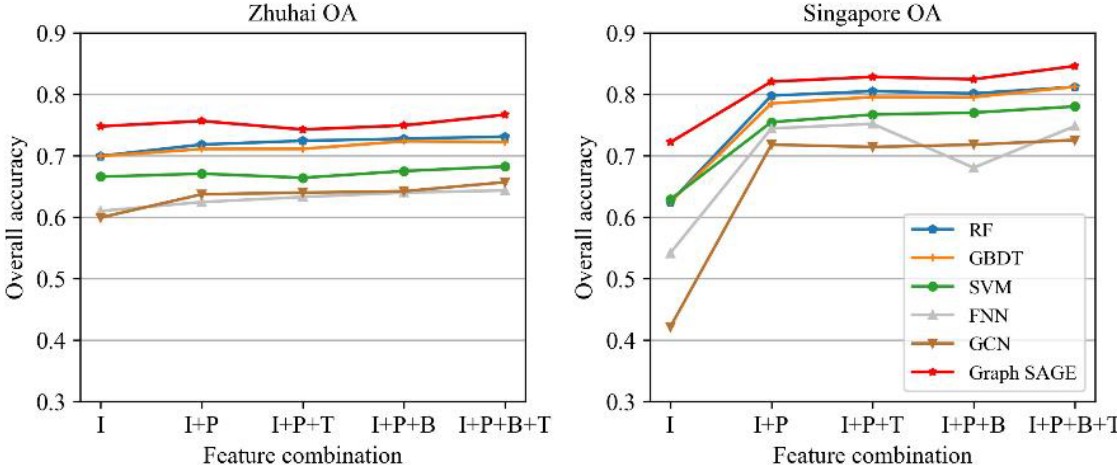

**Figure 15.** Overall accuracy of different models.

*5.3. Limitations of the Proposed Framework*

Although overall satisfactory results were achieved, limitations still exist in our proposed framework. First, we only constructed mobility graphs based on the spatial information extracted from the taxi trajectory data. The temporal features of mobility data should be considered in quantifying the mobility characteristics of UFZs. Second, the network structures of deep convolutional networks and graph convolutional models used in our study can be improved as more up-to-date deep learning network structures are introduced in UFZ classification tasks. The model tuning and structure adjustment should be further explored. The approaches to transforming the geographical objects into graph entities need systematic analysis with a large volume of geospatial datasets. Third, feature selection was not considered in our experiment. The identification of critical features that influence the UFZ mapping can be performed to extend our understanding of the influential factors on urban functional dynamics from the aspects of urban landscapes, socioeconomic attributes, and human activities. Fourth, the computational complexity of the proposed graph-based method is larger than common deep learning models and traditional machine learning methods.

## 6. Conclusions

This paper proposes the UFZ mapping framework based on graph-based classification. The physical features and social properties of functional zones are portrayed using multisource data, and a mobility graph is established to represent the mobility patterns of human movements. The graph-based models are used to classify UFZs based on a directed graph with semantic features. The experimental results in Zhuhai and Singapore demonstrate the effectiveness of the proposed graph-based classification method.

Specifically, this study contributes to three aspects. First, this paper demonstrates the efficiency of the proposed graph-based UFZ classification method, with an accuracy of 75.8% in Zhuhai and 84.5% in Singapore using multisource metrics. The proposed graph-based framework successfully exploits multisource features and achieves higher classification accuracies than traditional classification methods.

Second, trajectories reflecting human mobility patterns were introduced and applied to UFZ classifications. The functional zones are converted into a traffic network, and each UFZ is assigned with initial embeddings generated from multisource features. In contrast to existing studies applying undirected graphs to distinguish urban functions, our method integrated topological connections of urban blocks and human movements on the identification of UFZs. The evaluations prove that human mobility patterns can assist in identifying the functions of different urban zones.

Finally, the research framework can be easily generalized and applied to other classification scenarios. The experiments can be performed in other study areas and on datasets with only a portion of data sources. Furthermore, we can transfer the method to scenarios that utilize remote sensing images and social media data to achieve satisfactory UFZ classification results.

Our future work will include three aspects. First, we will incorporate temporal features of mobility data in the mobility graph to assist in identifying UFZs. Second, we will further explore the percentages of different functions within UFZ in order that the actual function of UFZ can be reflected more accurately. Third, we will establish an end-to-end framework that includes automatic feature selection.

**Author Contributions:** Conceptualization, C.F., Z.G., and J.W.; methodology, C.F., Z.G., and J.W.; processing and analysis of data, J.W.; visualization, J.W.; writing—original draft preparation, J.W.; writing—review and editing, C.F., Z.G., and J.W.; supervision, C.F. and Z.G.; funding acquisition, C.F. and Z.G. All authors have read and agreed to the published version of the manuscript.

**Funding:** This research is supported by the Ministry of Education, Singapore, grant No. FY2019-FRC4-008.

**Data Availability Statement:** The data presented in this study are available on request from the corresponding author.

**Acknowledgments:** The authors would like to thank the editor and the anonymous reviewers for their valuable comments, which helped improve this manuscript significantly.

**Conflicts of Interest:** The authors declare no conflict of interest.

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
