# Peer review of "A Novel Graph-Based Framework for Classifying Urban Functional Zones with Multisource Data and Human Mobility Patterns"

_remotesensing, doi:10.3390/rs15030730_

Round 1
Reviewer 1 Report
In this manuscript, the authors proposed a novel graph-based framework for classifying urban functional zones. The structure of the paper was good, and I was excited to read it. However, I still have some doubts about the paper.
1. The authors should give a clear definition and criteria of classification of UFZs as they were related to ontology.
2. In part 3.1, the authors used the OSM data as the main road network. In the result of Figure 8, it seemed that the road network was not fine enough. Is it possible to combine the OSM with high resolution images(0.6m) to provide a more detailed road network?
3. Could you please explain the relationships between building height and UFZ category?
4. In line 265, could you provide more information about the land use maps?
5. In part 5.2 table 10, what software or programming language did the author employ to conduct RF, GBDT,SVM,FNN..? In addition to the comparison of the OA, did the authors compare the time cost?
I hope these comments can further improve the manuscript.
Reviewer 2 Report
The paper proposed a graph-based UFZ classification framework that fuses HSR remote sensing images, points of interest, and GPS trajectory data, which is verified in two regions and achieved good results. The novel idea of this paper provides a new idea for urban functional zones classification. Some suggestions and questions about this article are as follows:
1. The article used the trajectory data of taxi and shared bicycle to build the graph, but there is a problem that not all UFZs have the travel data among them. If not, they will not be placed in the graph as a node and its function cannot be identified. How does the author deal with this part of UFZs?
2. The lable of UFZs in the paper are obtained by referencing for the land use data, but in fact the basic unit of land use data should not be the UFZ, and there are some inconsistencies in their basic unit. How does the author determine the actual function of UFZ? What are the criteria?
In addition, some UFZs include various types of ground objects, such as residential buildings, public service facilities, commercial centers and green spaces, which is a comprehensive urban functional zone. However, the article does not define the comprehensive UFZ. How does the author consider this issue?
3. In part 3.2.3, the author only mentions the 2D and 3D indicators and their calculation methods, but does not explain why these indicators are chosen, such as the building structural ratio and the tree height. What is the relationship between these indicators and UFZ types? It is suggested that the author can supplement this part.
4. When constructing the graph in part 3.3, the author mentioned that frequency of trips between two UFZs was recorded, but did not mention whether the frequency was used in the paper? Does the author take the frequency as a property of the directed edge of graph? If not, why record the frequency?
5. It is recommended to write part 3.3 in sections to make it more clear and readable. For example, 3.3.1 Construction of mobility graph and 3.3.2 the classification method for UFZs. This problem also exists elsewhere in the article, such as part 3.2.1.
Reviewer 3 Report
This paper proposed a graph-based UFZ classification framework that fuses semantic features from high spatial resolution (HSR) remote sensing images, points of interest, and GPS trajectory data. The paper is well-written but some problem still exist. The detailed comments are as follows.
1) Two many sources of data increase the difficulty of collecting data. Please clarify the practical potentials of the proposed method when not all types of data is available.
2) The rules that merge 14 classes into 6 categories should be explained since it decreases the difficulty of classifying urban functional zones.
3) There exist some grammatical mistakes, please polish the language.
4) Please compare the proposed method with state-of-the-art CNN based approaches to show the superiority.
